# Psychometric Properties of the Pittsburgh Sleep Quality Index (PSQI) in Patients with Multiple Sclerosis: Factor Structure, Reliability, Correlates, and Discrimination

**DOI:** 10.3390/jcm11072037

**Published:** 2022-04-05

**Authors:** Ana Jerković, Una Mikac, Meri Matijaca, Vana Košta, Ana Ćurković Katić, Krešimir Dolić, Igor Vujović, Joško Šoda, Zoran Đogaš, Sanda Pavelin, Maja Rogić Vidaković

**Affiliations:** 1Laboratory for Human and Experimental Neurophysiology (LAHEN), Department of Neuroscience, School of Medicine, University of Split, 21000 Split, Croatia; anasuto@gmail.com (A.J.); zdogas@mefst.hr (Z.Đ.); 2Department of Psychology, Faculty of Humanities and Social Sciences, University of Zagreb, 10000 Zagreb, Croatia; umikac@ffzg.hr; 3Department of Neurology, University Hospital of Split, 21000 Split, Croatia; meri.matijaca@gmail.com (M.M.); vanakosta@gmail.com (V.K.); ana.curkovic.katic@gmail.com (A.Ć.K.); spavelin@gmail.com (S.P.); 4Department of Radiology, University Hospital of Split, 21000 Split, Croatia; kdolic79@gmail.com; 5Department of Marine Electrical Engineering and Information Technologies, Faculty of Maritime Studies, University of Split, 21000 Split, Croatia; ivujovic@pfst.hr (I.V.); jsoda@pfst.hr (J.Š.); 6Sleep Medical Center, University of Split, 21000 Split, Croatia

**Keywords:** Pittsburgh Sleep Quality Index (PSQI), multiple sclerosis, sleep, validation, reliability

## Abstract

Sleep disturbances and poor sleep are a common complaint in the population with multiple sclerosis (MS) disease. The most commonly reported scale is the Pittsburgh Sleep Quality Index (PSQI), measuring seven components of sleep quality. Yet, till today, the PSQI instrument has not been validated in people with multiple sclerosis (pwMS). The objective of our study was to add precision in sleep quality assessment by investigating the psychometric properties of PSQI (factor structure, reliability, validity based on relations with other variables, cut-off scores) in pwMS. The cross-sectional study included data on a total of 87 patients with MS and 216 control subjects. Demographic information, education level, and MS-related variables were ascertained. Psychometric properties were examined by estimating the validity, including factor structure, metric invariance, and relations with other MS- and non-MS-related variables, reliability, and discrimination ability of the PSQI. The Croatian version of the PSQI had a two-factor structure which demonstrated loading and partial intercept invariance between pwMS and the control group. The global score and both subscales had high internal consistencies (McDonald’s omega and Cronbach’s alpha coefficients) in pwMS and showed expected relations with demographic and MS-related variables. PwMS differed significantly in the PSQI global score from the control groups, although receiver operating characteristics (ROC) curve analysis did not indicate a clear cut-off point. The PSQI is a reliable and valid scale and can be applied in clinical settings for assessing sleep quality in pwMS.

## 1. Introduction

Multiple sclerosis (MS) is a chronic autoimmune and inflammatory disease of the central nervous system with a broad and complex clinical picture affecting approximately 2–144 per 100,000 people in Japan, America, and Europe [1,2]. Clinical symptoms of MS include disturbances in motor functions, sensory deficits, visual impairments, fatigue, and cognitive and sleep disturbances. Poor sleep has been reported in people with MS (pwMS), with a prevalence of more than 50% of patients with MS having sleep disturbances [3,4]. Sleep disturbances in pwMS can be caused by the MS disease (demyelination affecting the suprachiasmatic nucleus) [5] or due to contributing factors often seen in pwMS (such as fatigue, pain, spasticity, anxiety, stress, depression) [3]. Poor sleep quality has been shown to be an independent predictor of reduced quality of life in pwMS [6]. Additionally, a substantial proportion (37–56%) of pwMS are at elevated risk of sleep disorders such as obstructive sleep apnea [7]. The screening inventory, such as the STOP-Bang questionnaire, has been validated to assess risks for obstructive sleep apnea in pwMS [7].

Sleep quality is often assessed using the Pittsburgh Sleep Quality Index (PSQI), a self-administered questionnaire used to evaluate sleep quality during the past month [8]. The validity of the PSQI has been confirmed by several studies in different patient populations and languages [9,10,11,12,13,14,15], but has not yet been in pwMS even though the PSQI is often used in MS research [16,17,18,19]. Analyses of seven components of PSQI (subjective sleep quality, sleep onset latency, sleep duration, sleep efficiency, sleep disturbance, use of hypnotic medication, daytime dysfunction) showed that increased sleep latency was the most frequent complaint in pwMS (especially in women), followed by sleep disturbance and daytime dysfunction [6,20]. The clinical course of MS may be influenced by sleep quality [16,18]. Given the importance of sleep quality in pwMS for understanding patient health and safety, the study goal was to examine the psychometric properties of the PSQI questionnaire for the first time in pwMS. Psychometric properties of PSQI were examined by estimating the validity, including the structure and relations with other variables, reliability, and discrimination ability. The factor structure was explored and compared between pwMS and people without MS. The study examined the associations of the PSQI with demographic variables and MS-related variables, such as the Multiple Sclerosis Impact Scale-29 (MSIS-29) [21,22]. The potential of PSQI to discriminate between pwMS and people without MS in terms of statistically significant differences and the receiver operating characteristics (ROC) curve and the area under the curve (AUC) was also explored, both in our sample and compared to data in previously published studies.

## 2. Materials and Methods

### 2.1. General Procedures

PwMS and control adult subjects were included in the study. The pwMS were recruited from the Department of Neurology of the University Hospital of Split (*n* = 70) and the Association of Multiple Sclerosis Societies of Croatia (*n* = 17). Inclusion criteria were as follows: (1) age 18 or older; (2) fluent in Croatian; (3) able to provide informed consent to all procedures; exclusion criteria were: (1) history of neurological disorder other than MS; (2) history of psychiatric disorder; (3) history of the developmental disorder (e.g., learning disability).

Since the study was conducted during the COVID-19 disease, the questionnaires were sent via Google survey. Eleven participants who were either older or not using technology completed the paper version of the questionnaires.

### 2.2. Participants

The participants recruited in the study were 87 pwMS and 216 control subjects. The data of all the control subjects were used when performing confirmatory factor analysis (CFA). When analyses were focused on the comparison, data of a control group (*n* = 134) composed of all women control subjects, and a random subsample of men chosen from the control subjects were used to achieve matching gender proportions with the pwMS group. The comparison of basic demographic characteristics for all three groups is shown in Table 1. The pwMS, control subjects, and the control group did not differ in any of the demographics, except for working status and comorbidities frequency. There were also differences in the type of comorbidities. The most common comorbidities for pwMS were endocrine, nutritional, and metabolic diseases (6.9%) and diseases of the respiratory system (4.6%). For control subjects, the most common comorbidities were diseases of the circulatory system (6%), endocrine, nutritional and metabolic diseases (3.7%), and musculoskeletal disorders (3.7%). The most common medications among the control subjects were antihypertensives (8.8%), pain medications (3.24%), and medicines for regulating thyroid function (2.78%).

Most MS subjects have been diagnosed with MS disease for 6 to 11 years (42.5%), 32.2% of people with MS have been diagnosed between 0 to 5 years ago, and 25.3% reported over 11 years of MS diagnosis. The mean duration of the disease for all people with MS was 9.14 years (*SD* = 7.290). A majority of the subjects had relapsing-remitting MS (RRMS) (79.3%), while others reported having primary progressive MS (PPMS) (9.2%) and secondary progressive MS (SPMS) (2.3%). Certain people with MS (9.2%) did not provide information on the type of MS. The median Expanded Disability Status Scale (EDSS) score for pwMS was 1, with 9% having a score of 4.5 or higher. Of 87 pwMS, 28.7% had comorbidities, including endocrine, nutritional, and metabolic diseases (6.9%) and diseases of the respiratory system (4.6%). Three pwMS were in wheelchairs. Of total pwMS, 58.62% were taking immunomodulatory or immunosuppressive drugs, most often glatiramer-acetate—Copaxone, i.e., Remurel (16.1%). The most common medications besides immune-related medications were vitamins and supplements (24.14%), thyroid hormones (9.2%), and pain medications (6.9%).

Thirty percent of pwMS and 7% of control subjects did not answer all questions. Each analysis was calculated on all the non-missing data for that analysis.

### 2.3. Questionnaires

Demographic data (age, sex, handedness), education level, comorbidity, and medication intake related to comorbidities were collected for all subjects. PwMS also reported MS-related information, including duration of the disease, MS types, EDSS score, and information on medication intake related to MS treatment.

#### 2.3.1. Pittsburgh Sleep Quality Index (PSQI)

The Pittsburgh Sleep Quality Index (PSQI)[8] is the most commonly used instrument for the subjective assessment of sleep quality in clinical and non-clinical populations. The questionnaire consists of 24 items, 19 of which are self-report and 5 of which are assessed by a sleeping partner if available and are not used when calculating the scores. Most of the items are evaluated on a 4-point scale, while four items require a numeric answer. The results are calculated according to a specific scoring key as seven component scores: sleep quality, sleep latency, sleep duration, sleep efficiency, sleep disturbance, sleep medication, and daytime sleep dysfunction. Authors originally suggested calculating the global PSQI score as the sum of all seven components, although more recent research does not agree if the one-factor structure is fully supported by data [23,24]. The validity of the PSQI has been confirmed by several studies in different patient populations [9,10] and in other languages [9,11,12,13,14]. In the present study, the Croatian translation of the PSQI was used [15].

#### 2.3.2. Multiple Sclerosis Impact Scale (MSIS-29)

The Multiple Sclerosis Impact Scale (MSIS-29)[21] is a self-report scale measuring the psychological and physical impact of the MS disease on the patient. The scale is structured as two subscales, a 20-item scale for measuring physical impact and a 9-item scale for measuring the psychological impact of the disease in the past two weeks. For each statement, the participant chooses the number that best describes his/her condition on a five-point Likert scale (from 1 = not at all to 5 = extremely). The MSIS-29 score is generated by summing the scores independently for each subscale, with higher scores indicating a more severe disease burden.

### 2.4. Validation Procedure

The CFA was performed to check the factor structure, which was compared to previous research [8,23,24,25,26], and to test the measurement invariance of the PSQI between pwMS and the control subjects. Reliability of the established scales was indicated by internal consistency indicators and compared with previous research [27,28]. Validity based on relations to other variables was tested by correlating PSQI and various demographic and MS variables and comparing the means of pwMS and other groups of subjects established in the present study and previously published studies on PSQI [15,19,28,29,30,31,32,33]. Discrimination between pwMS and control subjects on the basis of PSQI was further explored by examining the receiver operating characteristics (ROC) curve [34] and testing different possible cut-off scores.

### 2.5. Data Analysis and Statistics

The factor structure of PSQI was explored with CFA on the whole sample (pwMS and control subjects). Diagonally weighted least squares estimation with robust standard errors and a mean and variance adjusted test statistic was used. The models were estimated with multigroup analysis simultaneously for pwMS and control subjects. For the best fitting structural model, the measurement invariance of that model was tested between these two groups. The fit of all the models was judged by the significance of scaled χ^2^, the size of RMSEA (Root Mean Square Error of Approximation), CFI (Comparative Fit Indicator) and SRMR (Standardized Root Mean Squared Residual). The model fits were compared based on the significance of the difference in χ^2^.

For the scores suggested by CFA and previous literature [8], reliability was estimated by calculating two internal consistency indicators, McDonald’s omega, and Cronbach’s alpha coefficient. A global score composed of all components and two subscale scores were formed as item sums. Skewness and kurtosis indicators indicated deviations from a normal distribution, although visual inspection indicated normal-like distributions for most variables. Spearman’s rank-order coefficients (ρ) were used for correlations. *t*-test and ANOVA were used when comparing groups, considering their robustness to smaller deviations from normality. Levene’s test was used to assess the assumption of the equality of variances between groups. The Tukey-Kramer HSD post hoc test was calculated when using multiple comparisons.

The relations of PSQI scores with other variables were explored, partly on the whole sample and partly only for pwMS. Then, the discrimination power of PSQI was tested by comparing the scores of pwMS and people without MS and by analyzing the ROC curve. The Φ index, index of union, and Younden index [35], as well as previous research [9], were used to determine the cut-off point which could best differentiate pwMS from the control group. Finally, the relations of PSQI scores with relevant MS-related variables were explored.

In all calculations, a *p*-value of <0.05 was considered statistically significant. Data analyses were performed using the software IBM SPSS Statistics for Windows 25 (Version 25.0; IBM Corp: Armonk, NY, USA) [36], lavaan: An R Package for Structural Equation Modeling [37], MVN package for assessing multivariate normality [38], semTools for structural equation modeling [39], and StatPages [40].

### 2.6. PSQI Structure Procedure

To establish the PSQI structure, four basic models were compared simultaneously in pwMS and control subjects (Table 2). The one-factor model in which all seven components loaded on the same factor did not have an acceptable fit to the data. The two-factor model, with sleep duration and sleep efficiency as one factor and other components as second, had a significantly better fit to the data and fitted the data acceptably. In the three-factor model, one factor was composed of sleep duration and sleep efficiency, second of sleep disturbance and daytime dysfunction, and the third of subjective sleep quality, sleep onset latency, and use of hypnotic medication. Such a three-factor model fitted the data marginally better than the two-factor model (*p* = 0.047, Table 2). The inspection of parameters showed that the last two factors had a correlation 0.95 or higher in both groups, and the two-factor solution was therefore kept as the model best representing the data. In the bifactor model, all seven components loaded on one general factor, while sleep duration and sleep efficiency loaded on an independent specific factor, and other components on another independent specific factor. The bifactor model did not fit the data significantly better than the two-factor solution, prompting to keep the more parsimonious two-factor model as the best model. The two factors were PSQI sleep efficiency (composed of sleep duration and sleep efficiency components) and PSQI sleep quality (composed of other components).

Before testing the measurement invariance, some loadings were freed, which were kept equal in the previous analyses to ensure comparability with the one- and three-factor model. This led to a significant improvement in the fit to the data, Δχ^2^(2) = 6.55, *p* = 0.038, and therefore, this model was used when testing the invariance. Setting all the loadings to be equal across pwMS and the control subjects did not make the fit of the model significantly worse (Loading invariance 2F model in Table 2). However, setting all the intercepts to be invariant significantly worsened the model fit (Intercept invariance 2F model in Table 2). Then, the model was fitted in which all the intercepts were the same except for sleep efficiency (where the difference was largest), and this model fitted the data equally as well as the model with loading invariance (partial intercept invariance 2F in Table 2).

## 3. Results

### 3.1. PSQI Structure

The final two-factor structure of the PSQI is demonstrated in Figure 1. All the loadings, invariant across pwMS and control subjects, were significant and mostly high, and the correlation between factors was moderate in both groups. Based on this factor structure analysis, two subscores, PSQI sleep efficiency and PSQI sleep quality, were formed. Although the analysis strongly supported the expression of PSQI results as two subscores, the global score was calculated as is suggested in the first version of the instrument [8] and as is commonly used in research to ensure continuity and comparability.

### 3.2. PSQI Reliability

Table 3 presents McDonald’s omega as an indicator of PSQI sleep quality and efficiency subscores reliability (Table 3) [41]. The Cronbach’s alpha indicators are reported to enable comparison with other research (Table 3).

### 3.3. PSQI Relations with Other Variables

Women scored higher on the global PSQI and PSQI sleep quality but not on PSQI sleep efficiency (Table 4). Older people had higher results on all PSQI scores (Table 5). Table 5 also presents correlations of PSQI scores with MS-related variables: MSIS-29, EDSS, and duration of MS disease.

### 3.4. PSQI Discrimination

A *t*-test indicated pwMS have statistically higher PSQI scores than the control group (Table 6). One-way ANOVA indicated there is a significant difference in the mean scores on the global PSQI of pwMS and control group in the present study compared with means found in previous research (F(12, 2233) = 28.784, *p* < 0.001) [31,32,33]. Post hoc tests (Table 7) indicated that PSQI levels of the pwMS in the present study are similar to PSQI levels of pwMS in previously reported studies and are significantly different from the PSQI levels of almost all control groups in the present and previously reported studies [31,32,33]. The second ANOVA confirmed significant differences in mean PSQI scores between pwMS and the control group in the present study in comparison to mean PSQI scores of people suffering from sleep-related pathologies and people without pathologies reported in previous studies (F(5, 545) = 57.403, *p* < 0.001) [15,29]. Post hoc tests indicated that pwMS (M = 7.36, SD = 4.678) have lower PSQI scores compared to people with primary insomnia (M = 12.50, SD = 3.800) [29], do not differ from people with obstructive sleep apnea (M = 8.62, SD = 3.990) [15], and have higher scores than control groups in this and previous research (M = 3.3–5.6, SD = 1.8–3.1) [15,29].

**Table 6 jcm-11-02037-t006:** Differences in PSQI scores between pwMS (ms, *n* = 82–87) and the control group (con, *n* = 133–134).

Variable	*M*_ms_ (*SD*_ms_)	*M*_con_ (*SD*_con)_	Levene’s Test *F* (*p*)	*t* (*p*)
PSQI global score	7.36 (4.678)	5.60 (3.100)	29.74 (0.000)	3.08 (0.002)
PSQI sleep quality	5.65 (3.263)	4.16 (2.096)	21.68 (0.000)	3.72 (0.000)
PSQI sleep efficiency	1.96 (2.003)	1.45 (1.500)	14.56 (0.000)	2.01 (0.046)

**Table 7 jcm-11-02037-t007:** Significance (*p*) of Tukey–Kramer HSD post hoc tests of differences between groups of patients with multiple sclerosis (pwMS) and control groups (Con) in the present and previous research and their means and standard deviations.

Present Research			Lobentanz et al. [31]		Ma et al. [32]		Pinar et al. [33]	
			pwMS	Con	pwMS	Con	pwMS	Con
		*n*	388	991	231	265	50	50
	*n*	*M*	7.00 (3.900)	4.55 (3.710)	8.90 (5.200)	5.80 (4.800)	7.90 (3.500)	6.02 (3.220)
		(*SD*)						
		*M* (*SD*)						
pwMS	87	7.36 (4.678)	0.996	0.000 **	0.052	0.041 *	0.995	0.583
Con	134	5.60 (3.100)	0.014 *	0.090	0.000 **	1.000	0.015 *	0.999

*Note*. * *p* < 0.05. ** *p* < 0.01.

ROC analysis indicated that the global PSQI had AUC = 0.627, 95%CI [0.544, 0.710], *p* = 0.002 and PSQI sleep quality had AUC = 0.638, 95%CI [0.556, 0.719], *p* = 0.001, both of which were significant but indicate low accuracy in discrimination of these groups since they are close to 0.5 [34]. For PSQI sleep efficiency there was no significant discrimination, as indicated by AUC = 0.565, 95%CI [0.482, 0.648], *p* = 0.114. Index of union and Φ index suggested that the best cut-off point for discriminating between pwMS and the control group is score 6 and above on the global PSQI, and score 5 and above for PSQI sleep quality, while the Youden index suggests it is score 10 and above on the global PSQI, and score 7 and above for PSQI sleep quality. In Table 8, the proportions for participants categorized in groups of different PSQI levels according to these cut-off values are shown, as well as when using the cut-off of ≥5 suggested by Curcio et al. [9].

### 3.5. MS and PSQI

There were no significant differences in the global PSQI score nor in PSQI sleep quality between RRMS and people with other types of MS (all forming the same group due to a small number of people with other MS types (Table 9). However, people with RRMS had better PSQI sleep efficiency than other MS types (Table 9). Older people with MS had higher PSQI scores on all three PSQI variables than younger pwMS (ρ = 0.428 to 0.486, *p* < 0.001), similar to the whole sample analysis presented in Table 4. Unlike in the entire sample (Table 4), there were no gender differences (Table 9). There were no significant differences in any of the PSQI scores between pwMS married or cohabitating and pwMS single, separated, divorced, or widowed (Table 9). There were no differences between pwMS in sleep quality regarding the working status, but the groups differed significantly in the global PSQI score and the PSQI sleep efficiency (Table 9). Active (employed or student) and temporarily inactive (unemployed or temporarily on sick leave) pwMS had lower PSQI scores than permanently inactive pwMS (Table 9).

## 4. Discussion

The present study reported that the Croatian version of the PSQI had a two-factor structure which demonstrated loading and partial intercept invariance between pwMS and control subjects. We compared the fit of one-, two-, and three-factor models for PSQI proposed in previously reported studies [8,23,24,25,26]. In their systematic review, Manzar et al. [24] suggested multiple ways the two factors can be formed. In all of the two-factor solutions, the PSQI sleep duration and sleep efficiency were the common factor, while other components did not show such consistency. Based on this, we tested a two-factor solution, in which one factor was composed of sleep duration and efficiency and the other from the rest of the components, and this solution fitted the data better than the model with all components loading on the same factor (Table 2). In the systematic review of Manzar et al. [24], besides sleep duration and efficiency loading on the same factor, a common occurrence in the three-factor solutions was sleep disturbance and daytime dysfunction always being on the same factor, and therefore, we specified the second of the three factors as the combination of these two components. The third factor was composed of subjective sleep quality, sleep onset latency, and use of hypnotic medication, which was also supported by the frequency of such a factor in the overview of models by Manzar et al. [26]. The inconsistencies in factor solutions encountered in previous research [24,26] were also reflected in our results: the three-factor solution had a marginally better fit (Table 2), but the high correlation between second and third factors indicated the two-factor solution is better. The two-factor model is in agreement with the results of the systematic review that demonstrated that a two-factor solution was the most often obtained structure of PSQI [24].

The inconsistency in the structure in previous research might also indicate that PSQI has a bifactor structure, i.e., that all items load on the one general factor, but that they also load on specific group factors [42]. Bifactor structure would explain why in previously reported studies, some of the components loaded on a common factor and sometimes on specific factors. Because of this, the two-factor model was compared to a bifactor model with one general and two specific factors. However, the bifactor model did not prove to have a significantly better fit to data (Table 2), indicating there might not be one general factor explaining all the PSQI components. It might be more prudent to express PSQI as scores on two separate subscales, PSQI sleep efficiency and PSQI sleep quality, which have a medium to high correlation (Figure 1).

Based on measurement invariance testing, it was concluded that PSQI sleep quality measures the same construct in pwMS and control subjects and that both correlational and mean-level analyses comparing these two groups are justified. For PSQI sleep efficiency, only loading invariance was achieved, justifying correlation analysis, but intercepts were invariant for only one of two components, questioning whether mean-level comparison informs us on the difference in the construct or on the difference in the way the construct is related to its indicators.

To assess the reliability of the PSQI scores, two internal consistency indicators were calculated (Table 3). Although Cronbach alpha coefficient is one of the more used indicators of internal consistency, its estimation of reliability can be biased when the scale does not have a one-factor structure (i.e., when the items are not congeneric) and when the loadings on the common factor are not of the same size (i.e., when the items are not tau-equivalent) [41]. This is why it is not applicable for the PSQI global score given its two-factor structure, nor for the PSQI sleep quality given the differences between loadings (Figure 1). In such a case, McDonald’s omega can be considered a better indicator of internal consistency, although Cronbach’s alpha coefficient was also presented to ease comparison with previous research. In pwMS, the scores had levels of internal consistency similar to that in previous research [23], indicating high reliability. For control subjects, the consistency was somewhat lower than in the study of Cole et al. [23], although similar to Jia et al. [27] and Kmetec et al. [28], which might be a result of the smaller variance in the control subjects (Table 6).

The established gender and age differences in PSQI scores [27,28] were replicated in our sample (Table 4), indicating comparability of PSQI scores with previous research. Next, PSQI scores were expected to be higher (i.e., sleep quality lower) for people with the stronger psychological and physical impact of the MS disease on the patient [32]. This hypothesis was confirmed with the strong correlations of MSIS-29 subscales and PSQI scores (Table 5). As for EDSS, the predictions were not so precise: Lobentanz et al. [31] found higher EDSS is related to higher PSQI scores, Pinar et al.33 found no correlation and in Zhang et al. [19] the results were inconclusive. In the present research, the global PSQI and PSQI sleep efficiency had a significant correlation with EDSS, similar to Lobentanz et al. [31]. The correlation size was similar to Zhang et al. [19] (*r* = 0.248 vs. Cramer’s V = 0.205, χ^2^(6) = 11.926, *p* = 0.064, calculated based on the contingency tables presented by Zhang et al. [19]). PSQI sleep quality had an insignificant correlation, similar to Pinar et al. [33]. Regarding the MS diagnosis duration, most of the research did not establish the correlation with PSQI scores [19,31,33], which was also the case in the present study. However, the data presented in Zhang et al. [19] indicated that the relationship between MS diagnosis duration and sleep quality might not be linear, so we believe the research of this relation deserves further attention. The fact that two subscores, PSQI sleep quality and PSQI sleep efficiency, show different relations with other variables further supports the idea that PSQI should be used as two separate scores (Table 4 and Table 5).

Analyses of differences between pwMS and other groups indicated that pwMS on average showed lower sleep quality than control subjects, consistently in our and other research [15,29,31,32,33]. The sleep quality of pwMS is similar to that of people with obstructive sleep apnea and better than of people with primary insomnia [15,29]. However, different cut-off scores resulted in either a lot of pwMS being classified as not having low sleep quality or a large number of control subjects classified as having low sleep quality (Table 8). This is due to high variability in sleep quality in pwMS, with a similar number of subjects having low, moderate, or high PSQI scores. It might be that sleep quality is a problem for only certain subgroups of pwMS, e.g., only those with certain MS types or symptoms or with an inactive working status. These hypotheses need to be explored further on a larger sample to establish a clear cut-off point. Until then, we suggest using the generally accepted cut-off score 5 suggested by Curcio et al. [9].

To summarize, sleep disturbances and poor sleep are common complaints in the MS disease population. Yet, till today, the PSQI instrument has not been validated in pwMS. Our study demonstrated that the PSQI is a reliable and valid scale and can be applied in clinical settings for assessing sleep quality in pwMS. Sleep quality can influence the outcome of pwMS [16], and therefore, a reliable and validated tool is very useful for the quick screening of sleeping quality, as well as for evaluating the impact of medications on sleep profile in pwMS [18].

Our study has some limitations that need to be highlighted. The study limitation refers to the sample size with a larger number of men recruited in the control group. Further, the number of pwMS with different MS phenotypes (for progressive ones) was smaller in the present study, and therefore, exploring the influence of MS type on sleep quality and defining a clear cut-off point was not possible to investigate.

## 5. Conclusions

This is the first study that investigated psychometric characteristics of the PSQI, including factor structure, reliability, correlates, and discrimination in patients with multiple sclerosis. It also adds to the knowledge about PSQI in general by testing the models presented in previous reviews [24,26], including a new model of the bifactor structure, as well as exploring the measurement invariance between control subjects and pwMS. It would be recommended for future studies investigating sleep quality in pwMS to manage demographic, socioeconomic status, habits (i.e., smoking), physical activity, fatigue, anxiety, depression, quality of life, as well as cognitive aspects [43,44,45,46,47,48,49,50,51,52,53,54,55,56].

## Figures and Tables

**Figure 1 jcm-11-02037-f001:**
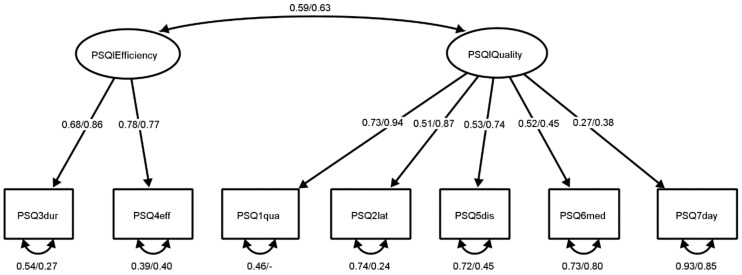
The two-factor model of PSQI scores for control subjects and pwMS. *Note*. Standardized parameters are presented for control subjects/pwMS. All reported parameters are significant at *p* < 0.05. PSQI components: PSQ3dur = sleep duration, PSQ4eff = sleep efficiency, PSQ1qua = sleep quality, PSQ2lat = sleep latency, PSQ5dis = sleep disturbance, PSQ6med = sleep medication, PSQ7day = daytime sleep dysfunction.

**Table 1 jcm-11-02037-t001:** Comparison of percent of participants with certain demographic characteristics in pwMS (*n* = 87), all control subjects (*n* = 216) and a subsampled control group (*n* = 134).

Variable	Category	pwMS	All Control Subjects	Control Group	pwMS—Control Group Comparison
Gender	Women	81.6	50.5	81.3	χ^2^(1) = 0.002 *p* = 0.960
	Men	18.4	49.5	18.7	
Right hand dominance	Yes	93.1	90.7	93.3	χ^2^(1) = 0.004 *p* = 0.951
	No	5.7	8.8	6	
Education	Primary school	3.4	2.8	3	χ^2^(3) = 5.453 *p* = 0.141
	High school	65.5	60.2	51.5	
	Undergraduate study	9.2	10.2	11.2	
	Graduate study	20.7	26.9	34.3	
Marriage status	Single	21.8	22.2	20.9	χ^2^(3) = 3.282 *p* = 0.350
	Marriage/cohabitation	63.2	68.1	71.6	
	Divorced/separated	10.3	7.5	4.5	
	Widow(er)	3.4	2.3	3	
Working status	Student	6.9	6	6.7	χ^2^(7) = 52.819 *p* < 0.001
	Employee	32.2	76.9	76.1	
	Unemployed	18.4	7.9	9.7	
	Temporary sick leave	6.9	0.5	0.7	
	Permanent incapacity for work	5.7	0	0	
	A person who runs the household	3.4	0	0	
	Disability pension	23	8.3	6.7	
	Other	2.3	0.5	0	
Comorbidity	No	64.4	80.1	78.4	χ^2^(1) = 4.663 *p* = 0.031
	Yes	28.7	15.3	17.2	
Age	M (SD)	42.57 (12.2)	43.8 (12.632)	43.78 (12.749)	*t*(219) = 0.700 *p* = 0.484
	Range	19–73	18–81	22–73	

*Note*. Comparisons between pwMS and all control subjects are not presented—they were of the same significance as between pwMS and the control group, except for gender, which had significantly different proportions due to the expected ratio of male to female MS patients in the general population.

**Table 2 jcm-11-02037-t002:** The fit of the different PSQI structure models and their differences.

Model	χ^2^ (*p*)	*df*	CFI	RMSEA (0% CI)	SRMR	Δχ^2^ (*p*)	Δ*df*
One-factor	75.83 (0.000)	32	0.862	0.097 [0.069, 0.126]	0.073		
Two-factor (2F)	46.03 (0.031)	30	0.949	0.061 [0.019, 0.094]	0.052	20.06 (<0.001)	2
Three-factor	34.45 (0.124)	26	0.973	0.047 [0.000, 0.086]	0.045	9.64 (0.047)	4
Bifactor	13.28 (0.774)	18	1.000	0.000 [0.000, 0.051]	0.022	20.33 (0.061)	12
2F with free loadings	29.16 (0.404)	28	0.996	0.017 [0.000, 0.067]	0.036	6.55 (0.038)	2
Loadings invariance 2F	40.62 (0.141)	32	0.973	0.043 [0.000, 0.079]	0.052	7.21 (0.125)	4
Partial intercept invariance 2F	44.33 (0.160)	36	0.974	0.040 [0.000, 0.075]	0.054	3.75 (0.441)	4
Intercept invariance 2F	58.44 (0.014)	37	0.932	0.063 [0.029, 0.093]	0.061	18.61 (<0.001)	1

*Note*. All the models were compared to the model above them, except bifactor and 2F with free loadings, which were compared to the two-factor model. CFI = Comparative Fit Indicator, RMSEA = Root Mean Square Error of Approximation, SRMR = Standardized Root Mean Squared Residual.

**Table 3 jcm-11-02037-t003:** Reliability of PSQI scores in pwMS (*n* = 79) and control subjects (*n* = 213) expressed as McDonald’s omega/Cronbach’s alpha.

Variable	pwMS	Control
PSQI global score	–/0.83	–/0.69
PSQI sleep quality	0.80/0.81	0.64/0.61
PSQI sleep efficiency	0.79/0.79	0.69/0.69

**Table 4 jcm-11-02037-t004:** Differences in PSQI scores between women (f, *n* = 175–180) and men (m, *n* = 121–123) regardless of MS diagnosis.

Variable	*M*_f_ (*SD*_f_)	*M*_m_ (*SD*_m_)	Levene’s Test *F* (*p*)	*t* (*p*)
PSQI global score	6.37(3.928)	5.08(3.243)	7.45(0.007)	3.12(0.002)
PSQI sleep quality	4.89(2.797)	3.57(2.125)	9.31(0.002)	4.61(0.000)
PSQI sleep efficiency	1.58(1.648)	1.57(1.671)	0.01(0.935)	0.09(0.930)

**Table 5 jcm-11-02037-t005:** Spearman’s rang correlations of PSQI scores with expanded disability status scale (EDSS), psychological (MSIS-29 psy) and physical (MSIS-29 phy) impact of the MS disease on the patient, and MS diagnosis duration (Duration).

Variable	Age ^a^	EDSS ^b^	MSIS-29 psy ^c^	MSIS-29 phy ^c^	Duration ^d^
PSQI global score	0.241 **	0.248 *	0.772 **	0.601 **	−0.135
PSQI sleep quality	0.148 *	0.084	0.826 **	0.664 **	−0.074
PSQI sleep efficiency	0.294 **	0.330 **	0.461 **	0.319 **	−0.143

*Note*. ^a^ *n* = 296–303; ^b^ *n* = 74–79; ^c^ *n* = 82–87; ^d^ *n* = 78–83. * *p* < 0.05. ** *p* < 0.01.

**Table 8 jcm-11-02037-t008:** Sensitivity (Sens), specificity (Spec), and proportions of patients with multiple sclerosis (pwMS) and the control group (Con) classified in the high/low PSQI group according to different cut-offs based on different criteria.

Variable	Cut-Off	Criteria	Sens	Spec	pwMS		Con	
					High PSQI	Low PSQI	High PSQI	Low PSQI
PSQI global score	5	Curcio et al.	0.671	0.429	0.644	0.356	0.575	0.425
	6	IuO/Φ	0.633	0.571	0.598	0.402	0.433	0.567
	10	Youden	0.392	0.895	0.368	0.632	0.104	0.896
PSQI sleep quality	5	IuO/Φ	0.595	0.609	0.590	0.410	0.391	0.609
	7	Youden	0.367	0.880	0.361	0.639	0.120	0.880

**Table 9 jcm-11-02037-t009:** Differences in PSQI scores regarding specific characteristics of pwMS.

Variable	Category	*n*	PSQI Global Score		PSQI Sleep Quality		PSQI Sleep Efficiency	
			*M* (*SD*)	*t*/*F* (*p*)	*M* (*SD*)	*t*/*F* (*p*)	*M* (*SD*)	*t*/*F* (*p*)
Type of MS	Relapsing-remitting	64–69	7.00 (4.462)	−1.40 (0.166)	5.61 (3.243)	−0.24 (0.808)	1.63 (1.768)	−2.58 (0.017)
	Other MS types	17–18	8.72 (5.345)		5.82 (3.432)		3.17 (2.358)	
Gender	Women	66–71	7.46 (4.601)	0.45 (0.651)	5.90 (3.299)	1.41 (0.163)	1.89 (1.890)	−0.54 (0.595)
	Men	16	6.88 (5.136)		4.63 (2.986)		2.25 (2.463)	
Marriage status	Married/cohabitating	53–55	7.02 (4.148)	−1.05 (0.298)	5.45 (2.978)	−0.73 (0.466)	1.70 (1.846)	−1.64 (0.105)
	Single/separated/widowed	29–31	8.19 (5.388)		6.00 (3.742)		2.45 (2.213)	
Working status	Active	33–34	6.12 (4.617)	6.39 (0.003)	4.82 (3.264)	2.69 (0.074)	1.41 (1.725)	8.36 (0.001)
	Temporarily inactive	19–22	6.27 (4.366)		5.30 (3.246)		1.32 (1.827)	
	Permanently inactive	24–25	9.92 (3.947)		6.72 (2.880)		3.25 (2.069)	

*Note*. To see if the working status of pwMS is related to their PSQI sleep scores, pwMS people were divided in three groups: active (employed or student), temporarily inactive (unemployed or temporarily on sick leave), and permanently inactive (permanent incapacity for work or disability pension).

## Data Availability

The data presented in this study are available on request from the corresponding author. The data are not publicly available due to privacy restrictions.

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
