# Peer review of "Psychometric Properties of the Pittsburgh Sleep Quality Index (PSQI) in Patients with Multiple Sclerosis: Factor Structure, Reliability, Correlates, and Discrimination"

_jcm, 2022, doi:10.3390/jcm11072037_

Round 1

Reviewer 1 Report

The method and results of the study are suitable for publication but minor revision is required.

1)In the Introduction section of the article, it was written that multiple sclerosis is associated with immune deficiency. This is not the correct term. MS is an autoimmune and inflammatory disease, not due to immune deficiency.

2) In the Introduction section, it was written that MS symptoms include vascular dysfunction and obesity. Obesity is a risk factor for MS and other inflammatory diseases but it is not a symptom for MS. Vascular dysfunction is not a symptom for MS also. 

Author Response

Author's Response to the Reviewer's comments is attached in the word document.

Reviewer 2 Report

Psychometric Properties of the Pittsburgh Sleep Quality Index 2 (PSQI) in Patients with Multiple Sclerosis: Factor Structure, Re-3 liability, Correlates and Discrimination

The Pittsburgh Sleep Quality Index (PSQI) is the most commonly used instrument  for the subjective assessment of sleep quality in clinical and non-clinical populations

 Sleep disturbances and poor sleep is a common complaint in the population with multiple sclerosis (MS) disease. Yet, till today, the PSQI instrument has not been validated in people with multiple sclerosis (pwMS). For this reason this paper is very interesting.

Infact the authors demonstreated that the PSQI is a reliable and valid scale and can be applied in clinical settings for assessing sleep quality in pwMS.

Slee disoreders are not only common, but important on the outcome of MS (add in discussion section this reference: Sleep quality can influence the outcome of patients with multiple sclerosis L Buratti 1, D E Iacobucci 2, G Viticchi 2, L Falsetti 3, S Lattanzi 2, A Pulcini 2, M Silvestrini 2) so have an easy and reliable tool for a quick screening is very useful

It is also useful for evaluating the impact of drugs on the sleep profile in MS (Add this reference:Sleep in multiple sclerosis patients treated with interferon beta: an actigraphic studyChiara Rocchi 1, Alessandra Pulcini 1, Cristina Vesprini 1, Viviana Totaro 1, Giovanna Viticchi 1, Lorenzo Falsetti 2, Maura Chiara Danni 1, Marco Bartolini 1, Mauro Silvestrini 1, Laura Buratti 1)

Author Response

Author's response to Reviewer's comments is attached in the word document.
